# Safety Evaluation of Remdesivir for COVID-19 Patients with eGFR < 30 mL/min without Renal Replacement Therapy in a Japanese Single-Center Study

**DOI:** 10.3390/healthcare10112299

**Published:** 2022-11-17

**Authors:** Takumi Umemura, Yoshikazu Mutoh, Takahito Mizuno, Mao Hagihara, Hideo Kato, Tetsuya Yamada, Yoshiaki Ikeda, Hiroshige Mikamo, Toshihiko Ichihara

**Affiliations:** 1Department of Infection and Prevention, Tosei General Hospital, Seto 489-8642, Japan; 2Department of Pharmacy, Tosei General Hospital, Seto 489-8642, Japan; 3Department of Clinical Infectious Diseases, Aichi Medical University, Nagakute 480-1195, Japan; 4College of Pharmacy, Kinjo Gakuin University, Nagoya 463-0021, Japan

**Keywords:** remdesivir, renal impairment, SARS-CoV-2, eGFR

## Abstract

There are limited reports on the safety of remdesivir for patients with severe kidney disease. We investigated the safety of remdesivir administration for COVID-19 patients with estimated glomerular filtration rate (eGFR) <30 mL/min. This single-center retrospective study was conducted between March 2020 and April 2022 at Tosei General Hospital, Japan. Propensity score matching was performed between patients with eGFR ≤ 30 mL/min and eGFR >30 mL/min with remdesivir administration. The primary outcome was 30-day mortality after the first administration. Adverse events, including development of acute kidney injury (AKI), liver function disorder, anemia, and thrombocytopenia 48 h after the end of remdesivir administration, were evaluated. After propensity score matching, 23 patients were selected from each group. There were no differences in the 30-day mortality (risk ratio [RR] 1.00; 95% confidence interval [CI] 0.18–5.56). Development of AKI and liver function disorder was not statistically different between the two groups (RR 1.05; 95% CI 0.96–1.14 and RR 0.48; 95% CI 0.04–5.66, respectively). There was no trend toward a significant increase in adverse events in the eGFR < 30 mL/min group and severe renal dysfunction had little effect on the safety of remdesivir treatment.

## 1. Introduction

Remdesivir (RDV) is one of several globally approved antiviral drugs for severe acute respiratory syndrome coronavirus-2 (SARS-CoV-2) [1,2]. The clinical impact of RDV use for coronavirus disease 2019 (COVID-19) has been demonstrated for its treatment and prevention in patients at high risk for progression to severe disease and is recommended for mild to severe COVID-19 [3,4]. In Japan, remdesivir has been administered as the VEKLURY^®^ injection, which contains remdesivir in the form of lyophilized powder and approximately 3 g of sulfobutylether-beta-cyclodextrin (SBECD) as a solubilizing excipient.

Administration of VEKLURY^®^ for patients with estimated glomerular filtration rate (eGFR) < 30 mL/min is not recommended since SBECD is renally cleared and accumulates in patients with decreased renal function [1,2]. In some clinical trials relating to the efficacy of RDV, patients with eGFR < 30 mL/min were excluded and the safety of its use in patients with impaired renal function has not been evaluated adequately [3,4,5]. In real world data, there are limited reports on the safety of RDV for patients with severe kidney disease [6,7]. It is crucial to clarify whether RDV can be safely used in patients with severe kidney disease as part of the therapeutic strategy in the treatment of COVID-19. Therefore, we investigated the safety of RDV administration in patients with COVID-19 with eGFR < 30 mL/min.

## 2. Materials and Methods

### 2.1. Settings

We conducted a single-center retrospective study using medical records dated from March 2020 to April 2022 at Tosei General Hospital (633 bedded hospital). The study was approved by the Ethics Committee of Tosei General Hospital (receipt No. 1062). Patients with COVID-19 who were administered RDV were included in this study. Patients aged <18 years and those receiving all renal replacement therapy were excluded. COVID-19-confirmed patients were defined as those with a positive reverse transcription polymerase chain reaction (RT-PCR) or rapid antigen test for SARS-CoV-2 from nasopharyngeal or salivary swabs, regardless of symptoms.

### 2.2. Data Collection

We collected clinical data on age, sex, body weight, severity of COVID-19, risk factors for progression to severe COVID-19, and clinical laboratory data. The severity of COVID-19 was defined using the World Health Organization classification [8] and risk factors included hypertension, cardiovascular or cerebrovascular disease, diabetes mellitus, obesity (body mass index, BMI; ≥30), immunodeficiency, chronic, mild, or moderate kidney disease, chronic liver disease, chronic lung disease, and current cancer, or sickle cell disease [4]. We divided patients into two groups, patients with eGFR ≤ 30 mL/min and those with eGFR > 30 mL/min at the time of admission, to determine whether severely impaired renal function increased the frequency of adverse events with the use of RDV.

### 2.3. Propensity Score Matching Analysis

Propensity score matching was performed with a logistic regression model to generate a propensity score for balancing the baseline characteristics and potential confounders between the two groups. Patients were matched one-to-one by propensity score using the covariates of age, sex, the severity of COVID-19, high risk for progression to severe COVID-19 (yes or no), and concomitant nephrotoxic drug usage [6,7]. Safety outcomes were evaluated between the eGFR ≤ 30 mL/min and eGFR > 30 mL/min groups. The 30-day mortality from the start of RDV administration and adverse events, such as development of acute kidney injury (AKI), liver function disorder, anemia, and thrombocytopenia, were evaluated 48 h after the end of RDV treatment [6,7]. AKI was defined as serum creatinine > 1.5 times the baseline level within 48 h after the end of RDV administration compared to the baseline value obtained on the first day of RDV administration [6,9]. Liver function disorder was defined as aspartate aminotransferase (AST) or alanine transaminase (ALT) elevation of more than five times (equivalent to grade 3 Common Terminology Criteria) the upper limit of normal within 48 h of RDV administration [6,10]. 

### 2.4. Statistical Analyses

Qualitative and stratified continuous variables were compared using the Fisher’s exact or Pearson’s χ^2^ tests. Continuous variables were compared using the Mann–Whitney U test. Statistical significance was set at *p* < 0.05. All analyses were performed using IBM SPSS Statistics, version 25 (IBM, Armonk, NY, USA).

## 3. Results

We enrolled 227 patients who were treated with RDV in this study. One patient < 18 years, four patients receiving hemodialysis, and 16 patients in whom eGFR could not be assessed were excluded. The eGFR < 30 mL/min group comprised 23 patients. After the propensity score matching, 23 patients were selected from each group, and the baseline characteristics were balanced between the two matched groups (Table 1). Administration periods of RDV were 5 (IQR 3.5–5) days in the eGFR < 30 mL/min group and 5 (IQR 5–5) days in eGFR ≥ 30 mL/min group (*p* = 0.62). Some patients received concomitant drug COVID-19 treatment alongside RDV in the eGFR < 30 mL/min group: nine patients received corticosteroids (eight dexamethasone and one methylprednisolone), three received tocilizumab, and two received baricitinib. Similarly, in the eGFR ≥ 30 mL/min group, 11 patients received corticosteroids (nine dexamethasone and two methylprednisolone), two received tocilizumab, and two received baricitinib.

Comparing the two groups, there was no difference in 30-day mortality (risk ratio (RR) 1.00; 95% confidence interval (CI) 0.18–5.56). There were three deaths in each group and the main causes of death are as follows: in eGFR < 30 mL/min group, two deaths were related to COVID-19 and one to bacterial pneumonia; in the eGFR ≥ 30 mL/min group, two deaths were related to COVID-19 and one to severe dehydration. With regards to adverse events, the development of AKI and liver function disorder were not statistically different between the two groups (RR 1.05; 95% CI 0.96–1.14 and odds ratio (OR) 0.48; 95% CI 0.04–5.66, respectively) (Table 2). Anemia was observed in two patients in the eGFR < 30 mL/min group (8.7%) and one patient in the eGFR ≥ 30 mL/min group (4.3%). Thrombocytopenia was observed in one patient in the of eGFR < 30 mL/min group (4.3%) and two patients in the eGFR ≥ 30 mL/min group (8.7%). Figure 1 shows the trend data of serum creatinine of patients in the eGFR < 30 mL/min group who received RDV treatment. Three patients died during the follow-up period. The serum creatinine levels of patients with eGFR < 30 mL/min showed no abrupt increase after RDV initiation and a slow decrease over time was noted. 

## 4. Discussion

These findings suggest the safe use of RDV for patients with eGFR < 30 mL/min. There are limited reports on drug administration to patients with eGFR < 30 mL/min. RDV and GS-441524, which is the predominant active metabolite, are excreted via the urine and reduced renal function will cause delayed elimination of GS-441524 [3,11]. However, the systemic effects of increased GS-441524 concentration remain unclear. Moreover, VEKLURY^®^ for injection contains RDV in the form of lyophilized powder and SBECD. Some reports have shown the clinical safety of SBECD in patients with mild or moderate renal impairment; however, no clear conclusion was made for those with severe renal impairment [12,13,14]. 

Our study in patients matched by propensity score showed no significant increase in the incidence of adverse reactions in either the eGFR < 30 mL/min or the eGFR ≥ 30 mL/min group. This is the first study to suggest the safe administration of RDV in Asian patients with eGFR < 30 mL/min. 

Ackley et al. reported the safety of RDV administration among 40 patients with severe renal function impairment, defined as an estimated creatinine clearance < 30 mL/min [6]. In the development of AKI, with ALT elevation and abnormal liver function tests as safety outcomes, there was no statistically significant increase in either of the groups. However, there was a higher incidence of 30-day mortality in their study for patients with severe renal impairment than in the other group (55.9% vs. 16.4%) [6]. Our results showed no significant difference in 30-day mortality between both groups. The reason for the difference in results might be because the severe impairment group in the other study was older and had a higher rate of ventilator use and vasopressor or inotrope use during RDV administration than the control group. In contrast, our data were appropriately confounder-adjusted for age and severity of illness. 

Moreover, Seethapathy et al. compared 40 patients with eGFR < 30 mL/min per 1.73 m^2^ treated with RDV against 40 patients with eGFR < 30 mL/min per 1.73 m^2^ who were not treated with RDV (historical control) and evaluated the efficacy and safety of RDV using propensity score matching [7]. In their results, one patient who was administered RDV developed worsening kidney function compared with three patients in the historical control group. They also found no increased risk of cardiac arrhythmia, cardiac arrest, altered mental status, clinically significant anemia, or liver function test abnormalities [7]. 

Considering our results and these reports, we found no evidence showing that the use of RDV might be harmful in patients with eGFR < 30 mL/min. This study had some limitations. First, our results were based on a retrospective review of one institution; hence, the number of cases was not sufficient, and more cases need to be accumulated to evaluate the safety profile. Second, our results were based on routine clinical data at a municipal hospital. Therefore, it was difficult to characterize the concentrations of remdesivir, its active metabolites (such as GS-441524), and SBECD. Moreover, the virus variant (such as alfa, beta, delta, and omicron) was not specified, and no adjustment was made for the impact of the virus variant on patient prognosis. Additionally, it is difficult to identify direct causes in adverse events. 

## 5. Conclusions

The presence or absence of severe renal dysfunction had little impact on the safety of RDV treatment in patients with COVID-19. There was no trend towards a significant increase in adverse events in the eGFR < 30 mL/min group. The present study could demonstrate an important role in the choice of COVD-19 treatment for patients with severe renal failure.

## Figures and Tables

**Figure 1 healthcare-10-02299-f001:**
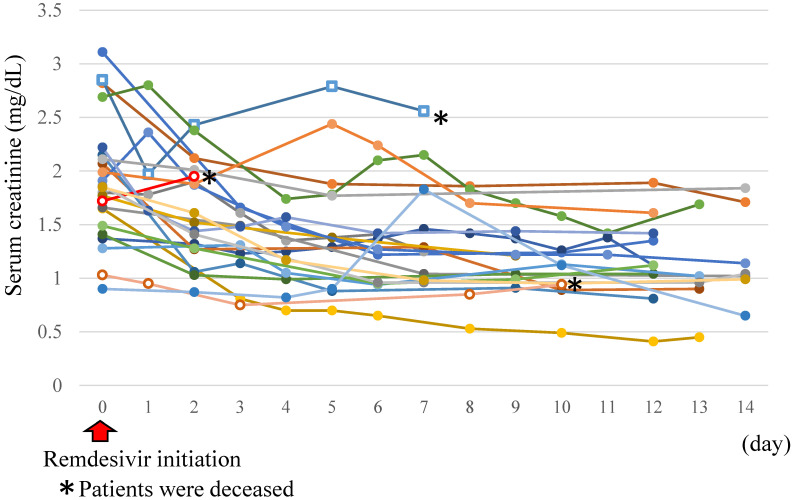
Trend data of serum creatinine in patients with eGFR < 30 mL/min with remdesivir treatment.

**Table 1 healthcare-10-02299-t001:** Patient characteristics pre- and post-matching.

Variables	Prematch			Postmatch		
	eGFR < 30 mL/min n = 23	eGFR ≥ 30 mL/min n = 183	*p* Value	eGFR < 30 mL/min n = 23	eGFR ≥ 30 mL/min n = 23	*p* Value
eGFR, median (IQR), mL/min	22.3 (18.7–26.3)	64.5 (47.7–78.2)	<0.01 ^a^	22.3 (18.7–26.3)	47.1 (45.0–50.0)	<0.01 ^a^
Age, median (IQR), years	80 (74–86)	72 (54–81)	0.02 ^a^	80 (74–86)	81 (78–88)	0.37 ^a^
Sex (male/female)	14/9	125/58	0.47 ^b^	14/9	11/12	0.38 ^b^
Severity of COVID-19						
Mild	11	47	0.14 ^b^	11	6	0.31 ^b^
Moderate	9	103		9	13	
Severe/critical	3	33		3	4	
Risk factor for progression to severe COVID-19 (%)	23 (100.0)	164 (89.6)	0.70 ^b^	23 (100.0)	23 (100.0)	1.00 ^b^
Age > 60 years	22	60		22	23	
Hypertension	10	64		10	8	
Cardiac disease	6	18		6	5	
Cancer	6	33		6	7	
Diabetes	5	47		5	8	
Chronic lung disease	5	39		5	7	
Chronic kidney disease	5	2		5	1	
Cerebrovascular disease	3	20		3	2	
Immunosuppression	3	20		3	3	
Smoking	3	73		3	3	
Dementia	2	16		2	8	
Obesity	0	24		0	2	
HIV	0	0		0	0	
Mental disorders	0	2		0	0	
Concomitant nephrotoxic drug usage (%)	5 (21.7)	18 (9.8)	0.10 ^b^	5	5	1.00 ^b^
TMP-SMX	2	0		2	0	
Loop diuretics	2	14		2	4	
Tacrolimus/cyclosporine	1	3		1	0	
Vancomycin	0	1		0	1	

^a^ Mann–Whitney U test, ^b^ Pearson’s χ^2^ test; eGFR: estimated glomerular filtration rate; IQR: interquartile range; COVID-19: coronavirus disease 2019; TMP-SMX: trimethoprim-sulfamethoxazole.

**Table 2 healthcare-10-02299-t002:** Safety outcomes of patients with remdesivir administration after propensity score matching analysis.

Outcomes	eGFR < 30 mL/min (n = 23)	eGFR ≥ 30 mL/min (n = 23)	*p* Value
30-days mortality (%)	13.0 (3/23)	13.0 (3/23)	1.00 ^a^
Acute kidney injury (%)	0 (0/23)	4.3 (1/23)	1.00 ^a^
Liver function disorder	8.7 (2/23)	4.3 (1/23)	1.00 ^a^
AST elevation	2	1	
ALT elevation	0	0	
Others			
Hyperglycemia	21.3 (5/23)	21.3 (5/23)	1.00 ^a^
Anemia	8.7 (2/23)	4.3 (1/23)	1.00 ^a^
Thrombocytopenia	4.3 (1/23)	8.7 (2/23)	1.00 ^a^

^a^ Fisher’s exact test. eGFR: estimated glomerular filtration rate; AST: aspartate aminotransferase; ALT: alanine transaminase.

## Data Availability

The data used in this study are available from the corresponding author upon reasonable request.

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
