# Peer review of "Safety Evaluation of Remdesivir for COVID-19 Patients with eGFR < 30 mL/min without Renal Replacement Therapy in a Japanese Single-Center Study"

_healthcare, 2022, doi:10.3390/healthcare10112299_

Round 1

Reviewer 1 Report (Previous Reviewer 1)

I am satisfied with the reviews. Thank you.

Author Response

no additional comment

Reviewer 2 Report (Previous Reviewer 3)

The Authors addressed my remarks adequatelly. I have no further comments.

Author Response

no additional comment

Reviewer 3 Report (New Reviewer)

This manuscript dealing with real world use of remdesivir is of interest to all clinicians that treat COVID-19 patients.  Several areas require further clarification by the authors and are listed below:

1. why was only 23 patients in each group used for the study?  What is the power of this design and can the authors make statistical claims based on 23 patients/group?  Authors need to justify. 

2.  The authors state that 9 patients received steroid therapy but only describe 8 patients.  Authors need to determine if it is 8 or 9 patients. 

3.  It would be of interest to have been able to characterize the remdesivir and active metabolite drug concentrations in plasma for the two groups as well as the SBECD.  Any reason this was not performed? 

Author Response

#Reviewer3

  1. why was only 23 patients in each group used for the study?  What is the power of this design and can the authors make statistical claims based on 23 patients/group?  Authors need to justify. 

RESPONSE: Thank you for pointing this out to us. For 30-day mortality, according to “Rule of Three”, the number of cases is adequate to verify a true difference, given our result of 13% in each group. However, a lack of detection power is undeniable with regard to other adverse events. We believe that the number of cases is not less than that in previous studies (references [6] and [7]).

The low number of cases has been mentioned under limitations in the Discussion section (Page 5 Lines 159-161).

  1. The authors state that 9 patients received steroid therapy but only describe 8 patients.  Authors need to determine if it is 8 or 9 patients. 

RESPONSE: Thank you for pointing this out to us. We apologize for the inaccuracy in the reported number of patients. We have revised Page 3 line 99 as below.

“9 patients received corticosteroids (8 dexamethasone and 1 methylprednisolone)”

  1. It would be of interest to have been able to characterize the remdesivir and active metabolite drug concentrations in plasma for the two groups as well as the SBECD.  Any reason this was not performed? 

RESPONSE: Thank you for your suggestion. Our results were based on a retrospective review based on routine clinical data at a municipal hospital. Therefore, it was difficult to characterize the remdesivir and active metabolite drug concentrations. We have therefore added the following sentences under limitations in the Discussion section (Page 5 Lines 163-165):

“Second, our results were based on routine clinical data at a municipal hospital. Therefore, it was difficult to characterize the concentrations of remdesivir, its active metabolites (such as GS-441524) and SBECD. ”

Round 2

Reviewer 3 Report (New Reviewer)

No further queries for this review.  Thanks.

This manuscript is a resubmission of an earlier submission. The following is a list of the peer review reports and author responses from that submission.

Round 1

Reviewer 1 Report

This paper presents a safety evaluation of remdesivir for COVID-19 patients with eGFR <30 mL/min. The authors enrolled 227 patients in one Japanese hospital. After PSM, the authors selected 23 patients from each group. The results show that RDV is safe for patients with GFR <30 mL/min.

The paper is well-written and organized.

Some questions to help to understand the work are:

  • Why n=183 in Table 1?
  • What is the cause of death of the patients? 
  • Was it related to RDV or CKD?
  • Is it possible to identify any other factor that can explain the results, such as a concomitant drug, for example?

Author Response

We thank you and the reviewers for your thoughtful suggestions and insights. The manuscript has benefited from these insightful suggestions. I look forward to working with you and the reviewers to move this manuscript closer to publication in the International Journal of Environmental Health Research.

・Why n=183 in Table 1?

Response: Thank you for your question. There were omissions in the description of excluded patients. We have added the sentences in the revised manuscript. (page 2, lines 89-91).

“One <18 years patient, 4 patients receiving hemodialysis, and 16 patients in which GFR could not be assessed were excluded.”

・What is the cause of death of the patients? Was it related to RDV or CKD?

Response: Thank you for your question. All six deaths were not related to RDV or renal failure. We have added the sentences in the revised manuscript. (page 3, lines 102-105).

“There were three deaths in each group, and the main causes were: in the eGFR <30 mL/min group: 2 were due to COVID-19, and one was related to bacterial pneumoniae. Similarly, in the GFR ≥30 mL/min group, 2 deaths were related to COVID-19, and one was due to severe dehydration.”

・Is it possible to identify any other factor that can explain the results, such as a concomitant drug, for example?

Response: Thank you for your question. Unfortunately, identifying the direct cause of the results is challenging because our study was retrospective. Therefore, we have added the sentences as a limitation to this study. (page 5, line 158).

“Additionally, it is challenging to identify direct causes of adverse events.”

Reviewer 2 Report

Dear authors, 

The use of Remdesivir in COVID-19 patients has been the subject of great debate and hence extensively researched in the past two years. However the present study has some important shortcomings in terms of methodology:

1. One of the outcomes is reported as risk of AKI and the cohort is composed of patients with significant CKD. Searching for AKI in a CKD patient is somehow strange. Maybe acute on chronic kidney failure? But there are no criteria stated for this. 

2. Statistical section should be detailed as there is no actual description of statistical tests used in order to assure appropriate statistical analysis. 

3. Table 1 - risk factors... in the GFR<30, why are there only 3 pts diagnosed with CKD? Are the rest AKI patients? If yes they should have been excluded as AKI is an outcome. If no... they all have CKD. 

4. Most importantly, I don't see a power analysis to justify the small cohort. At a glance, the study is underpowered for the stated outcomes. 

Author Response

We thank you and the reviewers for your thoughtful suggestions and insights. The manuscript has benefited from these insightful suggestions. I look forward to working with you and the reviewers to move this manuscript closer to publication in the International Journal of Environmental Health Research.

The manuscript has been rechecked, and the necessary changes have been made in accordance with the reviewers’ suggestions. Revisions in the revised manuscript are in yellow highlights. The responses to all the comments have been prepared and given below.

  1. One of the outcomes is reported as risk of AKI and the cohort is composed of patients with significant CKD. Searching for AKI in a CKD patient is somehow strange. Maybe acute on chronic kidney failure? But there are no criteria stated for this.

Response: Thank you for your comment. The literature we mentioned reported on AKI development in CKD patients, and previous studies investigated CKD individuals with AKI using the same criteria as those without CKD. (JAMA 2005;294(7):813-8. doi: 10.1001/jama.294.7.813). We, therefore, believe that our study is not an inappropriate approach.

  1. Statistical section should be detailed as there is no actual description of statistical tests used in order to assure appropriate statistical analysis.

Response: Thank you for this suggestion. The statistical tests used have been described in the Methods section and the footnotes of each table.

  1. Table 1 - risk factors... in the GFR<30, why are there only 3 pts diagnosed with CKD? Are the rest AKI patients? If yes they should have been excluded as AKI is an outcome. If no... they all have CKD.

Response: Thank you for your question. The current study classified patients according to their GFR at admission. Therefore, not all were patients with CKD. Similar studies have been previously conducted (reference 6); CKD definition is GFR <30 mL/min/1.73m, not GFR <30 mL/min. We have revised the Method section accordingly. (Page 2, line 65).

  1. Most importantly, I don't see a power analysis to justify the small cohort. At a glance, the study is underpowered for the stated outcomes.

Response: Thank you for pointing this out. Under-detection is undeniable; however, we believe that the number of cases is not inferior to previous studies. The low number of cases has been mentioned as a limitation in the Discussion section.

Reviewer 3 Report

The study addresses an important issue in the field of clinical pharmacology. Its cognitive value is limited by the small size of the study group and the lack of randomization, but the authors are aware of these weaknesses of the study and clearly indicate them as a limitation of the study. Hera are some detailed remarks:

1. The title of the study is misleading as it suggests that the study included patients with CKD stage 4 and 5, including those treated with renal replacement therapy. In turn, the description of the group shows that patients in the 5 stage receiving treatment with hemodialysis were excluded. I suggest to change the title of the paper so that it clearly describes the research group.

2. Settings, line 54: who was actually excluded from the study group? Only patients on hemodialysis or all patients on RRT? What about patients on peritoneal dialysis?

3. Please provide descriptive statistics on renal function (mean and range of GFR) in the compared groups. It's important to see how much the comparative groups differed in terms of kidney function. Giving only the threshold for subgroup division as 30 ml / minute does not explain this, and as a result, subjects with GFR slightly below 30 ml/min could be compared with subjects with GFR slightly above 30 ml/min.

Author Response

We thank you and the reviewers for your thoughtful suggestions and insights. The manuscript has benefited from these insightful suggestions. I look forward to working with you and the reviewers to move this manuscript closer to publication in the International Journal of Environmental Health Research.

The manuscript has been rechecked, and the necessary changes have been made in accordance with the reviewers’ suggestions. Revisions in the revised manuscript are in yellow highlights. The responses to all the comments have been prepared and given below.

  1. The title of the study is misleading as it suggests that the study included patients with CKD stage 4 and 5, including those treated with renal replacement therapy. In turn, the description of the group shows that patients in the 5 stage receiving treatment with hemodialysis were excluded. I suggest to change the title of the paper so that it clearly describes the research group.

Response: Thank you for your suggestion. We have changed the title accordingly:

“Safety evaluation of remdesivir for COVID-19 patients with eGFR <30 mL/min without renal replacement therapy in a Japanese single-center study”

  1. Settings, line 54: who was actually excluded from the study group? Only patients on hemodialysis or all patients on RRT? What about patients on peritoneal dialysis?

Response: Thank you for your question. We have excluded patients with renal replacement therapy and revised the sentences accordingly. (Page 2, line 53)

“Patients < 18 years and those receiving all renal replacement therapy were excluded.”

  1. Please provide descriptive statistics on renal function (mean and range of GFR) in the compared groups. It's important to see how much the comparative groups differed in terms of kidney function. Giving only the threshold for subgroup division as 30 ml / minute does not explain this, and as a result, subjects with GFR slightly below 30 ml/min could be compared with subjects with GFR slightly above 30 ml/min.

Response: Thank you for your suggestion. We have included the data in Table 1 as appropriate.

Round 2

Reviewer 2 Report

Dear authors, 

Unfortunately I don't consider that my concern have been addressed appropriately. Two main concerns still are present:

1. the diagnosis of AKI in patients with CKD. Just stating that another study did the same thing (study that was published more than 17 years ago) based on different diagnosis criteria does not make it medically correct. 

2. after describing the statistical analysis I was able to compute a power analysis and the study is well underpowered.